# A next-generation dual guide CRISPR system for genetic interaction library screening

Thomas Burgold [1], Emre Karakoc[1,2], Emanuel Gonçalves [1,3,4], Inigo Barrio-Hernandez[1,2,5], Lisa Dwane[1], Romina Silva [1,2], Emily Souster[1,2], Mamta Sharma[1], Alexandra Beck[1,2], Gene Ching Chiek Koh [1,6,7], Lykourgos-Panagiotis Zalmas[2], Mathew J. Garnett [1,2] & Andrew R. Bassett [1,2] ✉

Pairwise perturbation of gene function using the CRISPR/Cas9 system has potential in screening for genetic interactions and synthetic lethal gene pairs to identify combination therapies for cancer. However, existing dual guide expression systems are cumbersome to clone, often result in a large proportion of undesired guide pairs and have an imbalance of guide expression from the two positions. Here, we demonstrate a next-generation system for dual guide delivery based around a tRNA spacer that allows a single-step cloning strategy, as little as 2% of undesired guide pairs, and highly balanced expression of the two guides. This system allows efficient library-scale screening for hundreds of thousands of genetic interactions using the well-understood *Streptococcus pyogenes* Cas9 (SpCas9) system. We use this to screen a 100,136 guide pair library in colorectal cancer cells and successfully identify synthetic lethal genetic interactions between paralogs or other known interacting genes, establishing our method for performing efficient large-scale genetic interaction screens. This system is versatile and could be used with most guide RNA vector systems, and for other uses of paired guide delivery, such as improving single gene knockout efficiency or improving guide detection in single cell or optical CRISPR screens.

Genome-wide genetic screens for knockout using the CRISPR/Cas9 system have been immensely powerful in identifying genes that are essential for the viability of cancer cells[1,2]. These fitness genes are often good candidate targets for cancer therapy, but such whole-genome screens are limited to single perturbation phenotypes. Combination therapy is now the cornerstone of personalised medicine, and has been used in the clinic to reduce therapeutic side effects, enhance drug efficacy and critically, overcome drug resistance. However, the exhaustive number of possible combinations and the efficiency of screening methodologies have limited our ability to screen for and identify new combinations in an unbiased manner.

Current methods of delivery of guide pairs into a single cell often rely on the use of two independent pol III promoters, such as mouse and human U6 or H1[3–8]. This strategy has been used successfully, but has a number of limitations. Firstly, the use of two promoters often results in position-dependence of guide activity, such that the guide in one position is not as active as that in the second, increasing the number of controls that are necessary. Secondly, the length of the dual guide cassette makes libraries more difficult to clone, since the whole construct cannot be synthesised as an oligo pool, and can result in incorrect vectors due to the inefficiencies in each cloning step. Thirdly, the distance between the two guides is frequently hundreds of bases,

[1]Wellcome Sanger Institute, Wellcome Genome Campus, Hinxton, Cambridge, UK. [2]OpenTargets, Wellcome Genome Campus, Hinxton, Cambridge, UK. [3]Instituto Superior Técnico (IST), Universidade de Lisboa, Lisboa, Portugal. [4]INESC-ID, Lisboa, Portugal. [5]Instituto de Agrobiotecnología, IdAB, CSIC-Gobierno de Navarra, Mutilva, Spain. [6]Sir Jeffrey Cheah Sunway Medical School, Faculty of Medical and Life Sciences, 24 Sunway University, Sunway City, Malaysia. [7]Department of Genomic Medicine, University of Cambridge, Cambridge, UK. ✉e-mail: ab42@sanger.ac.uk

which results in recombinant molecules containing guide pairs that were not originally designed. This is due to a combination of template switching during PCR amplification and recombination during lentiviral production[9,10]. Whilst some of these limitations can be minimised by adjusting the orientation of the promoter-guide cassettes[11] or optimisation of library construction methods[8], it is still a significant proportion of the total.

An alternative strategy to avoid complex cloning involves the use of orthologous CRISPR enzymes, including paired SpCas9 and *Streptococcus aureus* Cas9 (SaCas9)[11] or *Acidaminococcus* (AsCas12a)[12,13]. The Cas12a system is particularly amenable to this, since several guide RNAs can be expressed as a single transcript that is processed by the Cas12a enzyme into individual units. This is therefore an attractive strategy for genetic interaction screening that has already been employed with some success[12,14], especially when more than two genes are knocked out simultaneously, which is difficult to achieve with Cas9-based systems. However, there are a number of limitations to this system. Firstly, even with the development of more efficient Cas12a enzymes and machine-learning-based guide design[15], the efficiency of mutagenesis is generally lower than for the more commonly used *Streptococcus pyogenes* Cas9 (SpCas9)[16]. Thus, two guides per gene are generally used for Cas12a screens. Since double-strand breaks are toxic, especially in primary cells, this makes genetic interaction screens using four guides more challenging. Secondly, because of its more widespread use, hundreds of cancer cell lines stably expressing Cas9 have already been made. Finally, Cas12a has a requirement for TTTV sequences as the protospacer adjacent motif[17] that can limit the targetability of certain genomic regions more than the NGG requirement for SpCas9.

Here, we develop and optimise a system to circumvent these limitations and allow efficient library-scale genetic interaction screens to be performed with SpCas9. We use a tRNA spacer between the two guide RNAs that is cleaved by the endogenous tRNA processing machinery at the RNA level, leaving the two guide RNAs. This is short enough, 193 base pairs, to allow guide pairs to be synthesised on a single oligonucleotide, thus allowing a single-step cloning into essentially any standard guide RNA vector. We develop this system to minimise intra- and inter-molecular recombination, reduce incorrect guide pair formation, and balance the activity of the guides in both positions. We demonstrate this technology by screening a library of over 100,000 genetic interaction pairs, and identifying synthetically lethal combinations of paralogs and other known genetic interactions in the HT-29 colorectal cancer cell line. Taken together, we present an optimised CRISPR/Cas9 genetic interaction technology, with several advantages over existing approaches, that allows efficient screening with hundreds of thousands of defined guide pairs.

## Results
### Establishment of tRNA-based dual guide system
In order to establish a system for expressing two guides that would be short enough to allow library-scale cloning of oligonucleotide pools, we exploited a tRNA-based spacer between the two guides. This has previously been shown to allow multiplexed guide expression in *Drosophila*, plant and human cells[18–20] and even for randomly paired genetic interaction screening (tRNA CRISPR for genetic interactions, TCGI)[21]. This exploits the highly abundant and efficient endogenous tRNA processing machinery that employs two RNAses (RNAse P and Z) to cleave at the 5′ and 3′ end of the tRNA in a structure-dependent manner. This is ideal for gRNA processing, since the cleavage is very precise, leaves no additional bases at either end, requires only a ~74 nt tRNA sequence, and is localised in the nucleus where the guide RNA molecules need to act. Additionally, the tRNA itself contains a pol III promoter that can support expression of the downstream sgRNA[22,23].

We assembled a plasmid containing the human glycine tRNA scaffold and 30 nt upstream leader sequence flanked by the two gRNAs

(Fig. 1A), and assessed the balance between the effectiveness of the guide in position 1 or position 2. We used two protospacer sequences, one targeting a *BFP* reporter gene that has been integrated at a single copy in the genome that we can assay by flow cytometry, and a second targeting the endogenous *SNCA* gene. We assayed mutagenic efficiency by high throughput amplicon sequencing across the CRISPR target sites (Fig. 1B). This demonstrated that the system was highly effective: the efficiency of mutagenesis was highly comparable (88–96% BFP, 92–96% SNCA editing) to single gRNA constructs (96-98% *BFP*, 94-96% *SNCA*) and independent of guide position within the vector (88-96% position 1, 88-96% position 2). These results were confirmed by flow cytometry analysis of *BFP* mutation, which showed similar results (76–95% editing, Supplementary Fig. 1A). In order to reduce the overall length of the construct to make oligo synthesis easier, we analysed the effect of reducing the length of the 30 nt leader sequence. This showed that a 6 nt leader had no effect on the efficiency (88-96% 30 nt, 89-96% 6 nt), but removing the leader altogether resulted in a partial loss of guide activity in the context of the upstream *BFP* guide (70% 0 nt, 95% 6 nt, Fig. 1B, Supplementary Fig. 1A).

### Library scale cloning of dual guides
In order to efficiently screen for genetic interactions or synthetic lethal combinations, it is necessary to make pooled libraries of many tens or hundreds of thousands of guide pairs. One of the advantages of the tRNA system is that the entire dual guide cassette consisting of two protospacers, guide scaffold and tRNA is only 193 nt in length, and even after adding cloning arms (244 nt), is short enough to be chemically synthesised as an oligo pool. This allows a one-step cloning of guide libraries into essentially any pre-existing gRNA expression vector by either Gibson assembly (Fig. 1C) or Golden Gate cloning.

In order to allow amplification of the library for cloning and sequencing of the guide pairs after screening, we also need to ensure there is a unique primer binding site that allows specific amplification of the guide pairs that is not shared in the first guide. This is made possible by utilising two different gRNA backbones, in this case the WT SpCas9 backbone in position 1 and a modified version with A-T base flip and extended stem loop[24] in position 2 (Fig. 1D).

In order to test and optimise library-scale cloning and screening, we designed a pilot library consisting of 8914 sgRNA pairs targeting 369 genes with a range of different effects on cell growth in the human colorectal adenocarcinoma cell line, HT-29, based on previous growth screens[1] (Supplementary Data 1). The library also included non-targeting gRNAs, guides targeting intergenic regions and core non-essential genes as negative controls. This provides a simple and sensitive readout of cell viability and we can test the sensitivity of the system with guides that are known to have a small or large effect size.

We included all guide pairs in both orientations to assess the positional bias. All single knockout control guides were paired with selected control guides targeting non-essential and intergenic regions as well as non-targeting guides, to understand which controls were the best and whether there was any bias introduced from making two double-strand breaks in the genome.

We used the human glycine tRNA sequence and 6 nt leader sequence in all constructs but also tested three different guide scaffolds in position 1 with a subset of the constructs (Fig. 1D). These were the wild type SpCas9 scaffold, and two mutated versions, one with a T to C flip in the lower stem to remove the pol III transcriptional terminator sequence (Mod6) and a second with additional substitutions in the nexus and hairpin regions (Mod7, Fig. 1D). These would potentially increase the efficacy of certain guides in position 1 but also reduce the chance of intramolecular recombination between the guide scaffolds.

We cloned this library into a lentiviral backbone[24] using Gibson Assembly, and analysed guide pair distribution by high-throughput paired-end sequencing. This showed that the representation of guides

within the library was very even (Fig. 1E, Supplementary Fig. S1B), with a Gini coefficient of 0.19, comparable to or better than single guide libraries (Yusa[24], Gini = 0.23) and better than other libraries made by two-step cloning (e.g., CDKO[25], Gini = 0.27). This evenness of guide coverage is an important factor to ensure high-quality library screens and reduce the number of cells needed through decreasing guide dropouts, increasing sensitivity and improving discrimination of essential genes[8,26]. The distribution of the cloned library was the same as that within the original oligo pool (Fig. 1E, Gini = 0.19), indicating that the cloning process had not introduced significant biases.

In previous methods of dual guide library generation, there are significant problems with template switching during PCR and recombination during lentiviral production and transduction that can result in guide pairs that were not designed or synthesised in the original library. This can amount to up to 50% of the total guide pairs in the library[9], and is highly dependent on the distance between the two guide pairs. One advantage of our tRNA-based system is that the distance between the two guide pairs is short (173 nt), which should reduce this problem. We thus looked for the proportion of the library that was made up of the desired or undesired guide pairs that are

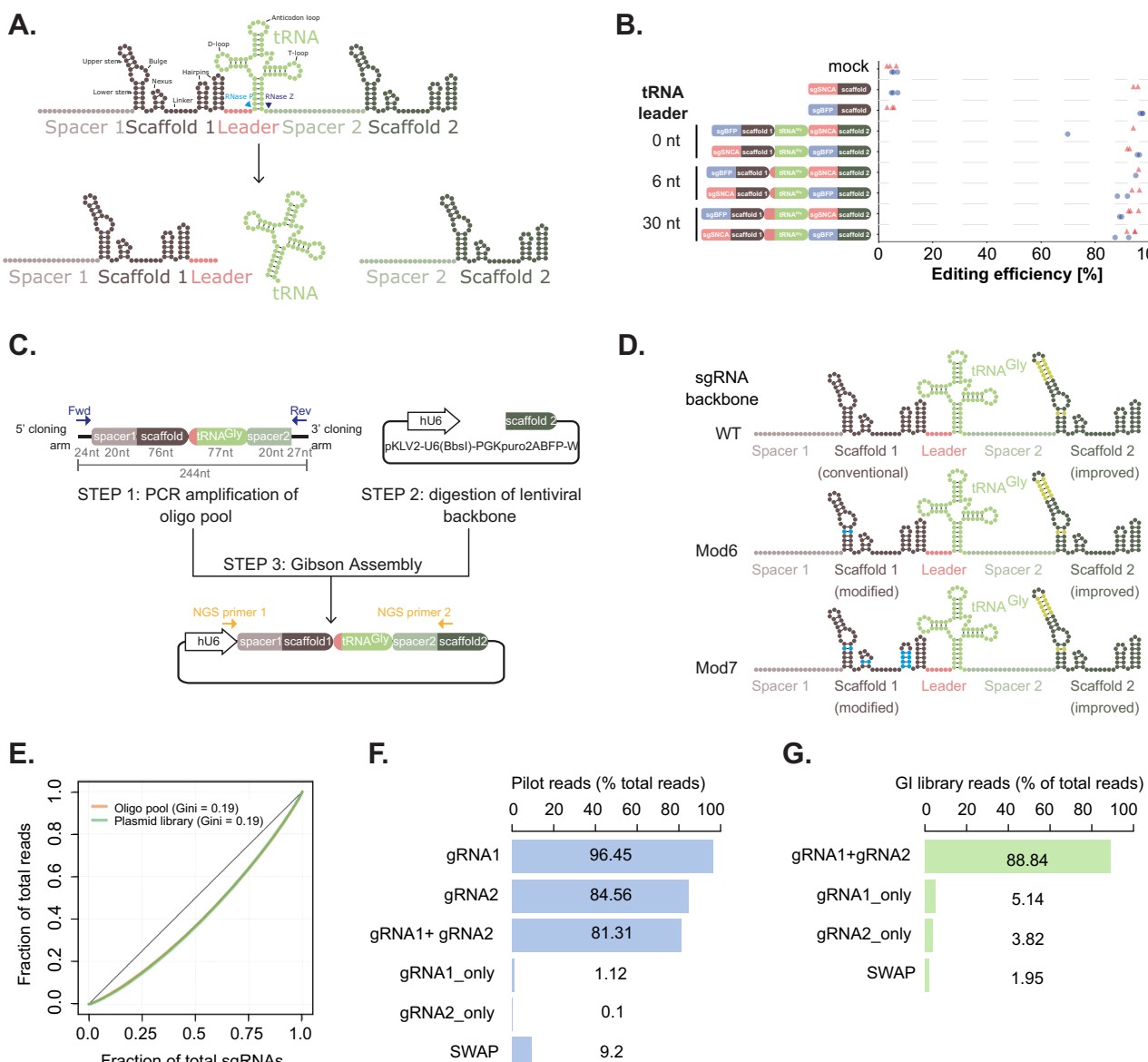

**Fig. 1 | Establishment of a dual guide system for pooled genetic interaction screening. A** Principle of the dual guide system. The two guide RNAs containing spacer and scaffold sequences are separated by a glycine tRNA preceded by a short leader sequence. The primary transcript is processed by RNAse P and Z in the nucleus to release the individual guide RNAs. **B** Editing efficiency of *BFP* (blue) and *SNCA* (red) guides in position 1 or 2 with different lengths of tRNA leader compared to single guide vectors. Data is shown as indel frequency assessed by high-throughput amplicon sequencing with individual biological repeats indicated. Source data are provided as a Source Data file. **C** Schematic of cloning of libraries of dual guide RNAs. Oligo pools are amplified by PCR and cloned by Gibson assembly into essentially any linearised guide vector. **D** Schematic of guide backbones in the dual guide cassette. WT is the original sgRNA scaffold[47], improved contains an extended stem loop and AT base flip[24], Mod6 has an AG base flip and Mod7 additionally contains other mutations throughout the scaffold sequence (see Methods). **E** Cumulative sgRNA distribution plot to analyse skew in the oligo pool (orange) or cloned plasmid library (green). Gini coefficients are indicated. **F** Proportion of reads in the pilot library with perfect matches to guide RNAs in position 1 (gRNA1), position 2 (gRNA2), both positions (gRNA1 + gRNA2), incorrect vectors with only one guide cloned from position 1 (gRNA1_only) or 2 (gRNA2_only) or guide pairs that were not designed in the library (SWAP). **G** as **F** but for the larger genetic interaction library, only showing both positions (gRNA1 + gRNA2), single guide vectors (gRNA1_only or gRNA2_only) and swaps (SWAP).

present in the original oligo pool, after cloning of the library and after lentiviral production and transduction of the library into HT-29 cells.

This showed that after PCR amplification and cloning of the oligo library, we observed 96.5% perfect mapping of gRNA1 and 84.6% mapping of gRNA2. There are greater numbers of sequencing errors in the second read, so gRNA2 mapping increases to 91.4% if we allow for mapping with one mismatch. Very few vectors had intramolecular recombination or incorrect cloning, resulting in a single gRNA1 or gRNA2 alone (1.1% or 0.1%). Importantly, we observed 81.3% of correct guide pairs mapping perfectly (rising to 87.1% if we allow for one mismatch), and only 9.2% where the pairing was incorrect. (Fig. 1F). These swaps could result from mistakes in the original oligo assembly, which at this stage was a pre-commercial product, or template switching during the PCR amplification, despite conditions being optimised to minimise this effect. These values were comparable between WT, mut6 or mut7 scaffolds (Supplementary Fig. 1C). The incorrect guide pairs did not show over-representation of specific pairs, suggesting that this was likely down to random recombination or template switching rather than a more systematic bias due to the methodology (Supplementary Fig. 1D).

Interestingly, when we analysed cloning of a larger library of over 100,000 guides (see below, Fig. 1G), we found the rate of undesired guide pairs was much lower, at only 1.95% in the cloned library, with 88.8% reads mapping to the designed guide pairs, whereas the levels of single guide vectors increased to 5.1% for sgRNA1 and 3.8% for gRNA2. The reduction in swap rates is likely to be due to developments in the oligo pool synthesis and cloning methodology, including a reduction in template concentration during the initial PCR to 20 pg/μl. Maintaining a low input DNA concentration of the oligo pool is critical to minimise template switching in the PCR[27] as well as minimising cycle number and increasing primer concentration and PCR extension time[10]. The lower swap rates could also result from a different composition of the library, which is an anchor-library design (Methods), meaning that there are fewer replicates of the same guide and that recombination events would more frequently recreate other guide pairs in the library. In conclusion, we show that our library construction methods are highly efficient, such that up to 88.8% of the plasmid library contains the correct guide pairs.

## Optimisation of screening with dual guide libraries

We next tested the functionality of our vectors by transducing HT-29 cells and analysing guide pair abundance after growth of cells for 3 and 14 days. Our library included a number of positive and negative controls, including guides that give varying degrees of effect on cell growth and viability. When we plotted the log2 fold changes in guide abundance between the day 3 and day 14 timepoints, we saw that, as expected, essential genes showed a significantly more negative log2 fold change than the negative controls (Fig. 2A, Cohen's D = 2.87 to 3.35, $p = 3.6 \times 10^{-237}$ to $2.4 \times 10^{-304}$). This is highly comparable to published single guide screens[1] in the same cell line (Supplementary Fig. 2A, Cohen's D = 3.16, p = $7.8 \times 10^{-62}$), demonstrating the effectiveness of the screen. This was highly consistent across repeat screens when performed at 500x or 100x coverage of the library (Supplementary Fig. 2B), as were the fold changes of all genes (Supplementary Fig. 2C).

We also correlated the effect size at the gene level observed in our screen with the results of a single guide vector system[1], which showed a strong correlation ($R^2 = 0.81$) and comparable effect sizes (gradient of regression line = 1.06), further confirming the effectiveness and reproducibility of our system (Fig. 2B). We note that direct comparison of effect sizes is difficult, as the composition of the libraries is different. In a genome-wide screen, most guides will have no effect on viability and thus the logFC is centred around zero. However, in our screen, many of the guides have an effect and thus non-essential genes have a logFC greater than zero (Fig. 2A) and the magnitude of the LFC for

essential genes would be expected to be less due to the normalisation used.

In order to PCR amplify the full-length dual guide cassette and minimise intramolecular recombination, we used three different guide scaffolds in position 1. Comparison of the effect size (Log2FC) of the wildtype scaffold (WT) and two modified forms (M6, M7) showed that the modified scaffolds had higher effect sizes and better separation of essential genes (Fig. 2A, Cohen's D WT = 2.87, M6 = 3.46, M7 = 3.34).

Another important issue with existing dual promoter vectors is an imbalance between the activity of the guides in position 1 and 2. In our pilot library, each guide was placed in both positions in the vector to measure this effect and thus we can plot the effect size (Log2FC) between the two positions (Fig. 2C). This showed that the activity of guides was highly correlated between the two positions across 630-804 different guides that have a range of effects on cell growth. We saw a slightly better correlation with less dispersion when using both of the modified scaffolds (Fig. 2C, Pearsons R for WT = 0.89, M6 = 0.93, M7 = 0.92). This is likely due to the removal of the pol III termination signal at the beginning of the guide scaffold that improves expression of guides, especially those whose target site contains multiple T bases at the 3' end[24]. Again, this was highly consistent across repeats and screens performed at 100x and 500x coverage of the library (Supplementary Fig. 2D). We thus decided to continue with the modified 7 (M7) scaffold, which is more different to the second scaffold in the vector in order to minimise intramolecular recombination.

We also analysed the recall curve of the known set of 246 essential guides in HT-29 established from single guide screens[1]. This showed that overall the recall of essential genes was very good (AUC 0.97-0.98, Fig. 2D) and highly comparable to the single guide screens (AUC 0.979, Fig. 2E). Using the WT scaffold in position 1, there was overall a high recall of essential genes with guides in the second position (AUC 0.99), but a slightly worse performance in the first position (AUC 0.96). Moving to either modified scaffold (M6, M7) removed this position dependence (Fig. 2D), resulting in comparable performance in the first position (AUC 0.98-1.0) and second positions (AUC 0.99-1.0). Results were independent of whether guides targeting essential genes were paired with non-targeting guides or those targeting intergenic regions. Again, these results were highly consistent between repeats and at 100x or 500x coverage of the library (Supplementary Fig. 2E, AUC 0.95-1.0). Using the plasmid library instead of day 3 as a control showed highly correlated log fold changes (Supplementary Fig. 2F) and recall of essential genes (AUC = 0.97-0.98, Supplementary Fig. 2G). Pairs of guides targeting non-essential genes or intergenic regions have a slight effect on cell viability (Fig. 2D, Supplementary Fig. 2E) when compared to non-targeting guides that do not cut the genome (AUC 0.02-0.05 for intergenic and non-essential compared to 0.00-0.01 for non-targeting guides). A similar effect has been observed in single guide experiments due to the toxicity of double-strand break formation, which has an effect on cell growth and viability. Thus, it is informative to control for the effect of double-strand break formation when designing controls for our libraries.

Together, this data demonstrates that our optimised system has highly comparable performance to previously published single guide screens[1] in terms of correlation of gene effect sizes (Pearsons R = 0.81), separation of essential and non-essential genes (Cohen's D = 3.34 versus 3.16) and recall of essential genes (AUC = 0.98-1.0 versus 0.979). The activity of guides is also essentially position independent (Pearsons R = 0.92). It also suggests that when we are designing control pairs to examine the phenotype of a single gene KO (singletons) for genetic interaction screens, we should pair these with guides that target the genome at intergenic sites rather than non-targeting guides to avoid overestimating the genetic interaction effects.

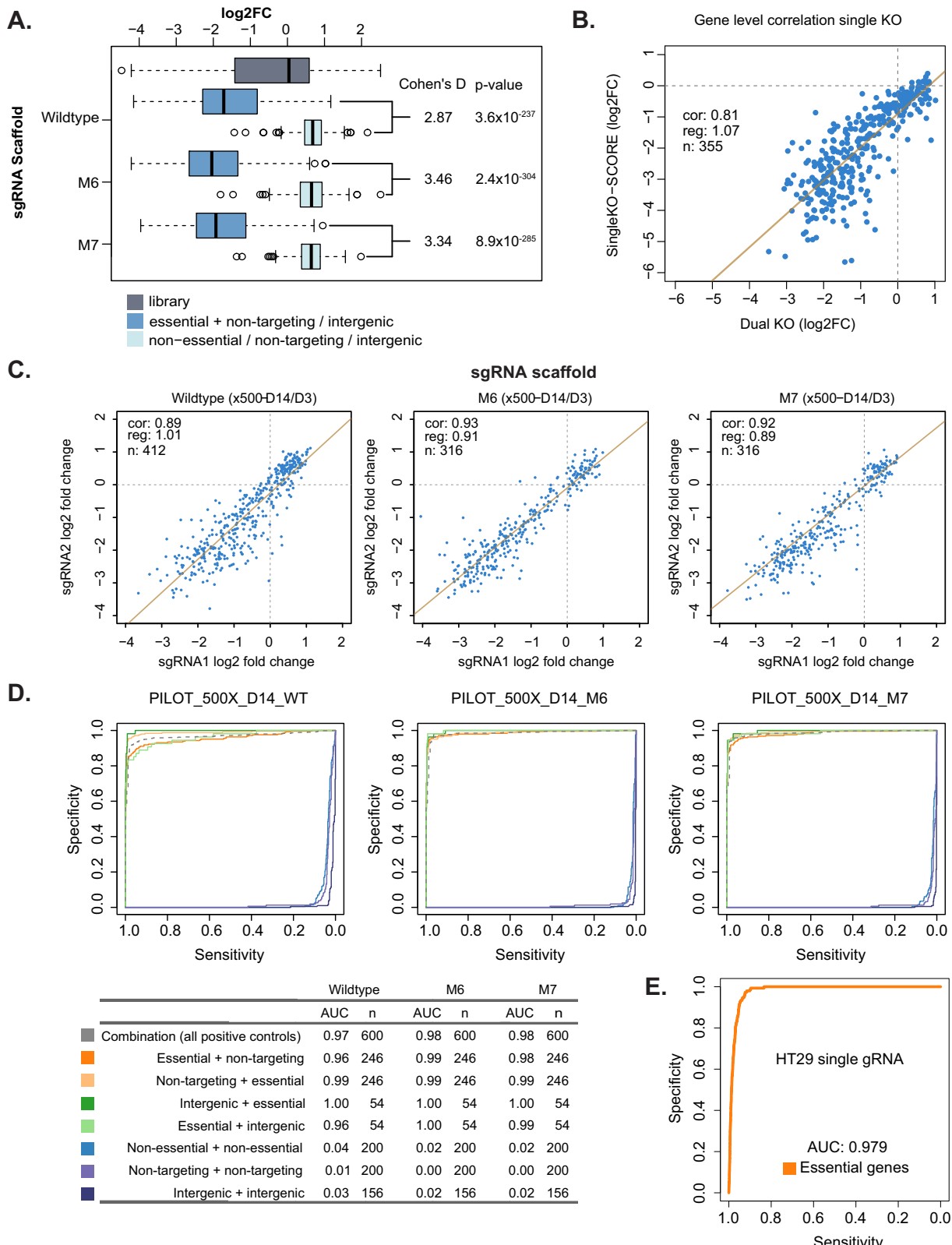

## Library scale screening with dual guide libraries

We next designed a large-scale library of 100,136 guide pairs to test the efficacy of our system in high-throughput screening for genetic interactions (see Methods). This was designed with 40 anchor genes that were each paired with 444 library genes, including all single gene controls (singletons) paired with guides targeting non-essential genes. We also included genetic interaction controls, including paralog pairs[11], known essential genes as positive controls and non-targeting, intergenic and non-essential genes as negative controls[1]. We used two guides targeting each gene, since we previously observed that this gave a good precision recall of known controls whilst enabling a dramatic reduction in library size and thus an increase in screening capacity[28]. Given the similarity in activity of guides in the two positions in the vector, anchor guides were always put in position 2, and library

**Fig. 2 | Pilot genetic interaction screen identifies optimal design. A** Boxplot showing the log2 fold change of sgRNA abundance for each gene between day 3 and day 14 (mean of three biological replicates). Data is grouped into guides targeting non-essential genes or intergenic regions (Non-essential/non-targeting/intergenic, $n = 556$ per scaffold), guides targeting essential genes paired with a non-targeting or intergenic guide (Essential + non-targeting/intergenic, $n = 600$ per scaffold) or the entire pilot library (Library, $n = 2970$). The effects with WT, Mod6 (M6) or Mod7 (M7) sgRNA scaffold in position 1 are shown. Box-and-whisker plots show interquartile range (box) 1.5x range (whisker) with outliers marked, centres indicate medians. Cohen's D values and two-sided t-test p-values are shown. **B** Correlation of the log2 fold change of sgRNA abundance per gene between day 3 and day 14 between dual guide vectors targeting one gene and an intergenic control and published data for single guide vectors targeting the same genes[1]. Each point shows the mean value of the guides for a gene and three biological replicates

($n = 355$). Linear regression is shown by a solid line and Pearson's correlation coefficient (cor), gradient of the regression line (reg) and number of genes ($n$) indicated. **C** Correlation of the log2 fold change of sgRNA abundance (mean of three biological replicates) with guides in position 1 (sgRNA1) and 2 (sgRNA2) separated by sgRNA scaffold in position 1 (WT, M6, M7). Linear regression is shown as a solid line with Pearson's correlation coefficient (cor), gradient (reg) and number of genes ($n$) indicated. **D** Recall curves of essential genes[1] depending on position within the vector and pairing with intergenic (green) or non-targeting (orange) guides. Different sgRNA scaffolds in position 1 (WT, M6, M7) are shown. Pairs of guides targeting non-essential genes (light blue) or intergenic (purple) regions are also compared to non-targeting (dark blue) guide pairs. Area under the recall curve (AUC) values for different combinations are indicated in along with number of genes used for these calculations ($n$). **E** Recall curve of essential genes ($n = 600$) from published data for single guide vectors[1] in the same cells.

guides in position 1 to further maximise the number of genetic interactions that could be screened.

We cloned the set of 100,136 guides in the same way as the pilot library, and analysis of the sequences by high-throughput sequencing showed that we had very even distributions of guide pairs (Supplementary Fig. 3A, gini = 0.21). The proportion of desired guide pairs in our library was very high (88.8%) and the percentage of undesired pairs very low (1.95%) with most of the remainder of the library being composed of single guides (8.96%) (Fig. 1G) and only a very small proportion (0.2%) of vectors with no match to guide sequences. We produced lentivirus from this plasmid pool, and used this to screen HT-29 cells in biological triplicate. As observed previously[9,29–31], there is recombination during the process of lentiviral production, which resulted in 3.82-3.88% of the library being incorrect guide pairs when transduced into Cas9 negative cells and 4.87-5.03% in Cas9 positive cells (Supplementary Fig. 3B). Single guide vectors also increased to 16.69-16.86% in Cas9 negative and 16.04-17.09% in Cas9 positive cells. However, given that we are sequencing both guides from the screened cells, we can unambiguously assign guide pairs, so this should not affect the precision of our results.

Comparison of the library sequencing at day 3 and day 14 after transduction into HT-29 cells showed that the internal essential gene controls were significantly depleted (Fig. 3A, Cohen's D = 2.05, $p = 1.1 \times 10^{-264}$) in comparison to the non-essential, intergenic and non-targeting controls, demonstrating the efficacy of the system. Similarly to the pilot library, the depletion of known essential genes correlated well with the results of published single guide libraries in the same cell type (Supplementary Fig. 3C, $n = 439$, $r^2 = 0.79$, gradient = 1.04), although we note again that due to differences in the composition of the libraries we cannot make direct comparisons in log-fold change (as for Fig. 2B). We also observed very good recall of essential genes using essential-intergenic or essential-non-targeting pairs in this library screen (AUC = 0.931, Supplementary Fig. 3D).

We calculated genetic interaction scores using a Bayesian methodology, GEMINI[32] for all 100,136 tested pairs. Often, closely related paralogous genes are able to compensate for the knockout of the other paralogous pair[33]. As expected, known paralogs had a significantly improved genetic interaction score ($p = 0.0124$, Fig. 3B). These include HDAC1 and HDAC2, MAPK1 and MAPK3, ASF1A and ASF1B and CNOT7 and CNOT8 (Fig. 3C), all of which have been previously validated in the literature[33]. These effects were highly consistent across multiple guides (Fig. 3C).

We also selected genetic interaction pairs from the BioGRID database, which have a significantly altered genetic interaction score in our screen ($n = 259$, $p = 0.01$), particularly when considering more confident pairs also backed up by protein-protein interaction data ($p = 0.0093$, $n = 61$, Fig. 3B). Several of these interactions were components of the same signalling pathway that show genetic interactions either due to partial redundancy or because they are acting as positive

and negative regulators. For example, we observed genetic interactions between BRAF and EGFR, which is important in the BRAF-mutant colorectal cancer cell line HT-29 (Fig. 3C). We also observe significant synthetic lethal interactions between BCL2L1 and MCL1, AKT1 and AKT2, EGFR and PIK3CA as well as EGFR and DMNT1 (Fig. 3C). Taken together, this validates the effectiveness of our screen in identifying genetic interactions.

## Validation of screen hits confirms paralog synthetic lethality

We validated two of the positive genetic interactions, CNOT7/8 and ASF1A/B in an arrayed manner, measuring cell growth over 6-7 days by imaging, and performing an endpoint cell viability assay. We cloned dual guide vectors with guides targeting each paralog coupled to a non-essential gene (CYP2A13) and vectors containing guides targeting both paralogs. This showed that knock out of individual paralogs had no or weak effects on cell growth (Fig. 3D) or viability (Supplementary Fig. 3E) when compared to untransduced cells. However, the combination of knockout of paralog pairs CNOT7/8 or ASF1A/B both resulted in significant reductions in cell growth (Fig. 3D) and endpoint viability (Supplementary Fig. 3E) of the cells, confirming the results of the pooled screen. Western blot analysis confirmed that protein was removed in the appropriate cell populations (Supplementary Fig. 3F, Supplementary Fig. 4).

In order to further validate that the effects we were observing were not an artefact of the dual guide vector system, we transduced two conventional single guide vectors targeting each gene. These were marked with two fluorescent markers such that by careful titration of the two viruses, we could generate a population of cells with single knockouts (marked by one or other fluorescent protein) or double knockouts (marked by presence of both fluorescent proteins) and analyse the proportion of these populations by flow cytometry over time to monitor the relative growth rates (Fig. 3E). By comparing day 3 and day 14 post-transduction, we could see a significant decrease in the double mutant population compared to the single knockouts for both CNOT7/8 and ASF1A/B pairs that we tested (Fig. 3E). In contrast, a non-essential pair (CYP2A13) did not show this effect. Together, these results validate the synthetic lethality of the CNOT7/8 and ASF1A/B paralog pairs and demonstrate the effectiveness of the dual guide system to identify genetic interactions.

## Discussion

We have developed a next-generation approach for genetic interaction screening based on a tRNA linker to deliver dual guide constructs that eliminates many of the problems with existing systems and enables library-scale screening of hundreds of thousands of interactions using CRISPR nucleases. We can clone libraries in a single step with even guide coverage, high efficiency and a low proportion of undesired guide pairs. Our system can be used with essentially any existing guide

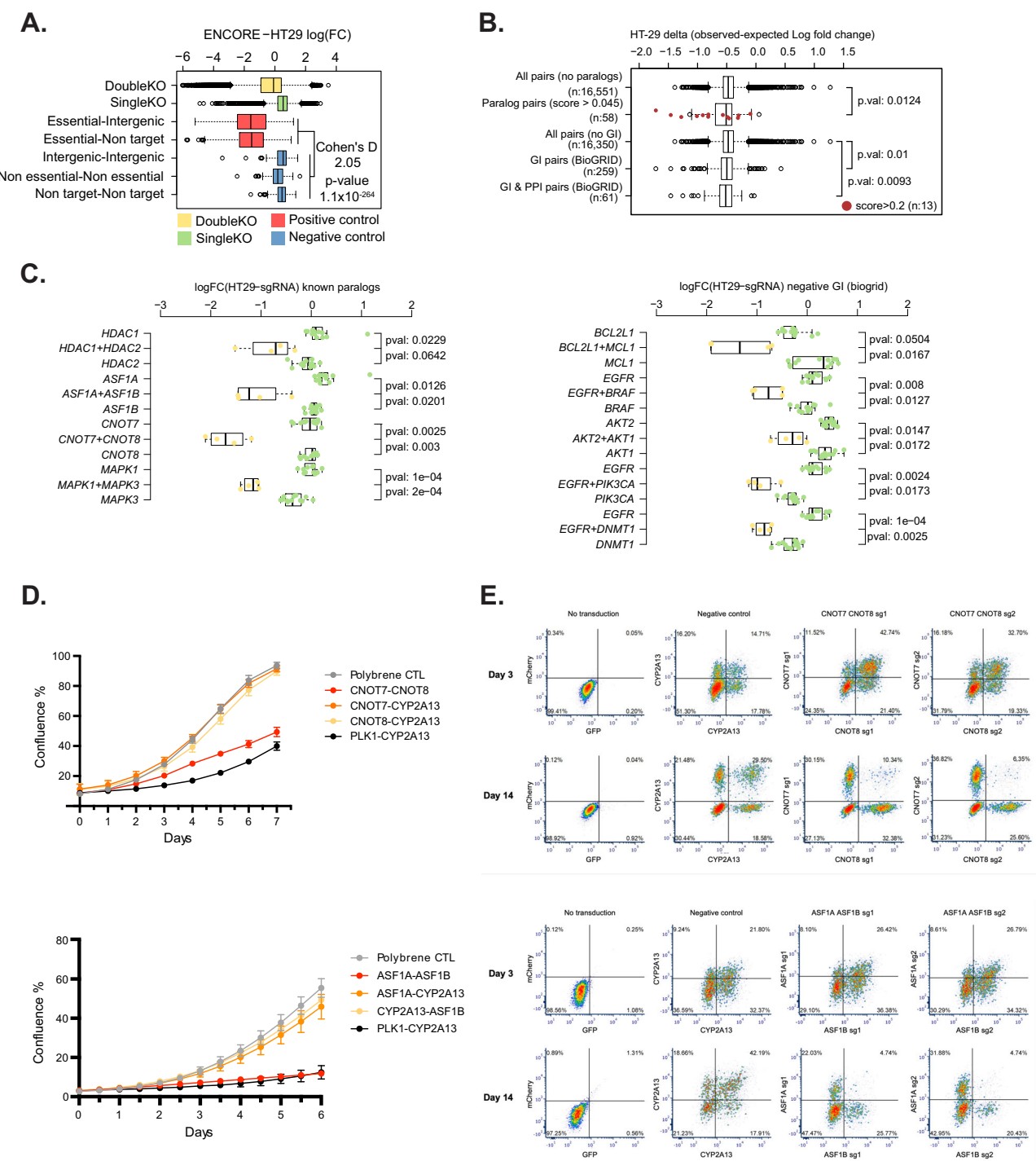

RNA vector, making it versatile, such as in primary cells where a single lentiviral vector delivering Cas9 and sgRNA is beneficial. Our methodology has been optimised to provide even guide activity between the two positions in the vector, and minimises intra- and inter-molecular recombination during the cloning and lentiviral production steps. Importantly, the use of *Streptococcus pyogenes* guide RNAs allows us to exploit the highly optimised systems, Cas9-expressing cell lines and considerable experimental data from previous screens to select guides[28] all of which maximise the efficiency of the screening. This in turn allows us to screen for a greater number of genetic interactions in a single experiment. We apply this to screening a 100,136 vector library and demonstrate its effectiveness in identifying synthetic lethal pairs such as gene paralogs and other known genetic interactions.

In the future, our methodology could be applied to delivery of guide pairs for other applications such as genetic deletion libraries[34] to identify functional non-coding regions of the genome, or to deliver pairs of prime editing and nicking guide RNAs into the same cell for prime editing screens of defined mutations[35]. Dual guide approaches will also make detection of guide RNAs in single cell[36,37] or pooled optical CRISPR screens[27] more sensitive, which is frequently limiting in primary or stem-cell derived cells. Our system is also amenable to improving single gene knockout by targeting the same gene with two guides to make a SuperMinLib[5,28] or improving CRISPR activation or inhibition efficiency[38,39] as has recently been demonstrated using a very similar system[27]. This will minimise the scale of genome-wide screening libraries to enable screens to be performed more simply in

**Fig. 3 | Large scale genetic interaction screening identifies paralogs and known synthetic lethal pairs. A** Boxplot showing log2 fold change in abundance of sgRNAs per gene between day 3 and day 14 (mean of three biological replicates). Data is grouped into non-targeting guides (non-target, n = 200), guides targeting non-essential genes (Non-essential, n = 200) or intergenic regions (Intergenic, n = 156), essential genes (Essential, n = 600), the entire library (DoubleKO n = 64,640) or single knockout controls (SingleKO, n = 5,808). Cohen's D and two-sided t-test are shown. **B** Boxplot showing genetic interaction values for paralogous pairs[33] (Paralog pairs) and non-paralogous pairs (All pairs no paralogs). Scores >0.2 are marked in red. Interacting pairs from the BioGRID database (GI pairs), those with protein-protein interaction evidence (GI & PPI pairs) or all others (All pairs no GI) are indicated, with two-sided t-test p-values. **C** Boxplots showing log2 fold changes in sgRNA abundance (mean of three biological replicates) for each paralog pair (left) or genetic interacting pair (right). Individual sgRNA pairs (dots) targeting one or other paralog coupled with control guides or both paralog pairs are indicated. Two-sided t-test p-values are shown. Boxplots in (**A**–**C**) show inter-quartile range (box) 1.5x range (whisker) with outliers marked, centres indicate medians. **D** Growth assays for *CNOT7/CNOT8* (upper panel) or *ASF1A/ASF1B* (lower panel). Untransduced cells (Polybrene CTRL, grey) are compared to a positive control targeting an essential and a non-essential gene (*PLK1-CYP2A13*, black) or combinations of each paralog coupled with a non-essential gene, *CYP2A13* (orange and yellow) or its paralogous pair (red). Error bars show standard deviation of three biological repeats, centres indicate mean values. Source data are provided as a Source Data file. **E** Flow cytometry plots of cells transduced with single sgRNA vectors targeting *CNOT7/8/CYP2A13* (upper panels) or *ASF1A/B/CYP2A13* at day 3 or day 14 post-transduction, marked with green fluorescence (x-axis) or red fluorescence (y-axis). Double transduction events contain cells with knockout of both genes. Two guides used for targeting (sg1 or sg2) are indicated. Loss of double positive cells at day 14 indicates synthetic lethality. Left panels show untransduced cells. Percentages of cells in each gate are indicated.

complex cell systems such as iPSC-derived cell types, xenografts, primary cells or organoids. Perhaps most importantly, we believe that our technology will enable simple, large-scale genetic interaction screening that will identify synthetic lethal combinations that are candidates for cancer therapy as well as mapping genetic interactions more broadly across the human genome.

## Methods

### Design of dual vector sequences
We designed the dual vector system following similar principles to Port et al.[18] with two sgRNAs separated by the human glycine tRNA. In order to have distinct sgRNA scaffolds in the two positions in the vector, we used the WT scaffold, or modified versions (Mod6 and Mod7 scaffolds) in position 1 coupled with the commonly used improved scaffold (imp) in position 2[24]. Both modified scaffolds contain a T-C substitution in position 5 and corresponding substitution in the other side of the stem loop to remove a TTTT polIII terminator sequence in order to improve expression. Mod7 additionally contains further substitutions that alter the sequence but maintain the secondary structure of the sgRNA.

**sgRNA scaffolds.** WT - gttttagagctagaaatagcaagttaaaataaggcta gtccgttatcaacttgaaaaagtggcaccgagtcggtgc

Mod6 - gtttcagagctagaaatagcaagttgaaataaggctagtccgttatcaactt gaaaaagtggcaccgagtcggtgc

Mod7 - gtttcagagctagaaatagcaagttgaaataagactagttcgttatcacgcc gaaaggcgggcaccgagtcggtgc

imp - gtttaagagctatgctggaaacagcatagcaagtttaaataaggctagtccgtt atcaacttgaaaaagtggcaccgagtcggtgc

**Human tRNA_Gly.** GCATTGGTGGTTCAGTGGTAGAATTCTCGCCTCCCA CGCGGGAGaCCCGGGTTCAATTCCCGGCCAATGCA

**Leader.** 30 nt - TGTTATCCTGCAGGCGGTTGTTACGCAGAG
6 nt - GCAGAG

The 30 nt tRNA leader sequence was taken from the transcript of the human glycine tRNA and reduced in length as described. All sgRNAs were 19 nt with the first base being a guanine to enable efficient transcription from the U6 promoter.

In order to test activity of the two guides, we employed sgRNAs targeting a single copy *BFP* transgene under the *EF1*-alpha promoter integrated at the *ROSA26* locus (target site CTCGTGACCACCTT-GAGCCA CGG) or the endogenous *SNCA* gene (target site GCAGCAG-GAAAGACAAAAGA GGG). The dual guide backbone1-tRNA cassette was synthesised (Gblock, IDT) and amplified with primers containing the two guideRNA target sites followed by Gibson cloning into pX458 (Addgene #48138). Plasmids were transfected into the KOLF2_C1 hiPSC line with TransIT-LT1 (Mirus Bio) and editing efficiencies measured at day 4 post-transfection by flow cytometry (gating on GFP positive cells,

and analysing the proportion of BFP positive and negative cells) or high throughput sequencing. Libraries for high throughput sequencing were made by PCR using KAPA HiFi HS ReadyMix (Roche) from genomic DNA (Qiagen blood and tissue kit) using primers BFP_F:atggtgagcaagggcga BFP_R:ggcatggacgagctgtacaag or SNCA_F: gctaatcagcaatttaaggctagc SNCA_R:cataggaatcttgaatactgggcc with tails to add on indexed sequencing adaptors for Illumina sequencing in a second round of PCR[40]. Samples were sequenced on an Illumina MiSeq instrument, and analysis performed using CRISPResso2[41].

### Design of pilot guide library
An initial dual-guide pilot library was assembled containing a total of 8914 vectors, each containing two sgRNAs, testing three different guide RNA scaffolds. The library was designed considering single-guide CRISPR-Cas9 screens obtained from the HT-29 cancer cell line selecting genes according to their essentiality in HT-29[1]. Selective and effective sgRNAs were selected using the same pipeline developed in Gonçalves et al.[28]. Six different classes of vectors were designed to test the library and consider specific technical aspects: 2952 vectors to test the position of the guides (Scaffold Position); 2872 vectors designed to target previously reported genetic interactions (GI pairs); 1884 vectors as negative controls, pairing non-essential or non-targeting sgRNAs (Negative Controls); 600 vectors targeting copy number amplified regions in HT-29, mostly focused on the chromosome 8 genetic region of *MYC* (Copy Number); 320 vectors targeting cancer related genes paired with non-targeting and intergenic sgRNAs; and 286 vectors were designed to test the insertion of double-strand breaks (DSB) at different distances. Each gene in the GI pairs has 4 sgRNAs and all combinations are considered, but guide order is fixed. A total of 200 non-targeting sgRNAs that had a uniform distribution and closer mean fold-change, when compared to a set of ~1000 sgRNAs that were screened[1], were selected. Intergenic sgRNAs with no mismatch alignments were selected from the LanderSabatini library[42]. Apart from Anchors and GI pairs (4 sgRNAs/gene) all other genes have 2 targeting sgRNAs. The library design choices were fully automated and implemented in the following script:

https://github.com/EmanuelGoncalves/crispy/blob/master/notebooks/dualguide/Library2Composition.py.

### Design of genetic interaction library
The large scale (100,136 vector) genetic interaction library paired 40 selected anchor genes with each of 444 library genes. Two optimal sgRNAs were selected for each gene from MinLibCas9[28], resulting in 4 possible guide pairs for each combination tested. We also included single gene knockout vectors where 4 guides targeting each anchor gene or 2 guides for each library gene were paired with each of 6 guides targeting three non-essential genes (*ADAD1,CYLC2* and *KLK12*). Details of the design and choice of genes and guides are explained in

more detail https://github.com/ibarrioh/DualGuide_COLO1/ and https://doi.org/10.5281/zenodo.17191951[43,44].

## Cloning of guide libraries

Oligo pools were synthesized by Twist Bioscience, resuspended in TE buffer (10 mM Tris-HCl, pH8.0, 0.1 mM EDTA) and amplified by PCR with Kapa HiFi Hotstart in 50 µl reactions with 1 ng library (10 ng for pilot library) using primers (IDT Ultramers, LibAmpF - ATCA-TATGCTTACCGTAACTTGAAAGTATTTCGATTTCTTGGCTTTATA-TATCTTGTGGAAAGGACGAAACACC.

LibAmpR - TGCCACTTTTTCAAGTTGATAACGGACTAGCCTTATT TAAACTTGCTATGCTGTTTCCAGCATAGCTCTTAAAC) and 14 cycles (Denaturation 95 °C 3 min, 98 °C 20 s, 71 °C 15 s, 72 °C 30 s, final extension 72 °C 60 s). Amplified products were purified with 0.7:1 ratio of AMPure XP beads:DNA and quantified by Qubit. 5 µg lentiviral guide expression vector pKLV2-U6gRNA5(BbsI)PGKpuro-2A-BFP-W (Addgene #67974) was linearised with 100 U of Bbs I-HF (NEB) for 2 h, and gel purified on a 0.6% agarose gel. Guide libraries were cloned into the backbone using Gibson assembly with 100 ng linearised vector and 12 ng insert in a 20 µl reaction volume at 50 °C for 1 h, followed by ethanol precipitation. A whole Gibson reaction (120 ng total DNA) was transformed by electroporation into Endura *E. coli* (Lucigen) and grown overnight in 500 ml LB media for 16 h at 32 °C. A serial dilution of the transformation mix was plated onto LB plates and used to estimate coverage of the libraries which was >1000 for both pilot and genetic interaction libraries. Up to 8 Gibson reactions were performed per library to achieve appropriate coverage.

## Virus production

293FT cells (Lonza) were transfected at 70-80% confluence in 10 cm dishes with 3 µg pKLV2 library plasmid, 7.4 µg psPAX2 and 1.6 µg pMD2.G plasmids in OptiMEM using 36 µl Lipofectamine LTX and 12 µl plus reagent. Virus particles were collected 3 d post transfection ali-quotted and stored at −80 °C, and titre measured by transduction into HT-29 cells and sorting for BFP. A minimum of 500x cells per guide pair were transfected to maintain viral coverage.

## Screening of HT-29 cells

The Cas9-expressing, female HT-29 cancer cell line (RRID:CVCL_0320, https://cellmodelpassports.sanger.ac.uk/passports/SIDM00136, obtained from the Sanger Institute cell model resource, commercially available from the National Cancer Institute collection) were cultured at 37 °C/5%CO$_2$ in RPMI media, supplemented with 10% FBS, 1% pen-strep, 1% sodium pyruvate and 1% glucose. Additionally, cells were maintained with Blasticidin (20 µg/ml) to maintain Cas9 expression. Cells were screened with both pilot and GI libraries in biological triplicate. Cells were transduced with the appropriate volume of lentiviral-packaged library to give a 30% multiplicity of infection (MOI) at 100X or 500X coverage for the pilot screen, and 100X for the GI library. The volume of virus to use was determined based on the titration of different virus volumes in a 6-well plate and detection of BFP-fluorescence using flow cytometry. The number of cells transduced was tailored based on the size of the library (3 ×106 cells for pilot library; 33 ×106 cells for GI library). Transduction efficiency was determined after 72 h, based on BFP-expressing cells using flow cyto-metry. Cells were then selected with puromycin to deplete the wild-type population, and library-positive cells were redetermined after 48 h. If sufficient (>80%), cells were maintained in growth for 14 days, keeping at least five-times the library representation at each passage. On day 14, cells were pelleted and stored and −80 °C for DNA extraction.

## Library preparation for high-throughput sequencing

Genomic DNA was extracted from cell pellets with the Qiagen blood and tissue kit, and PCRs performed using a total amount of DNA appropriate for 100-500x coverage of the library size (e.g., 100k library x 100 coverage = 10 M cells at ~5 pg DNA per cell = 50 µg genomic DNA). This was split into 3 µg per 50 µl PCR reaction using ExTaq HS enzyme and primers (IDT Ultramers, U1 - ACACTCTTTCCCTA-CACGACGCTCTTCCGATCTCTTGTGGAAAGGACGAAACA, R2 - TCGGCATTCCTGCTGAACCGCTCTTCCGATCTGCTGTTTCCAGCA-TAGCTCTT) to amplify the dual guide cassette with Illumina sequen-cing appends. 25 cycles of PCR were performed (Denaturation 95 °C 3 min, 98 °C 10 s, 64 °C 15 s, 72 °C 30 s, final extension 72 °C 60 s) and products purified with a Qiagen PCR purification kit followed by quantification with Qubit high sensitivity reagents. Illumina indexes were added by amplifying 1 ng DNA to a 50 µl PCR reaction using Kapa HS enzyme and unique dual indexing primers (Illumina) for 11 cycles (Denaturation 95 °C 3 min, 98 °C 20 s, 70 °C 15 s, 72 °C 20 s, final extension 72 °C 60 s). Samples were purified with 0.7:1 ratio of AMPure XP beads:DNA and quantified by Qubit and Tapestation. Libraries were sequenced on an Illumina MiSeq or NovaSeq using custom sequencing primers (R1 - U6-Illumina-seq2 - TCTTCCGATCTCTTGTGGAAAGGAC-GAAACACCG and R2 - iScaffold-Illumina-seq3 – GCTCTTCCGATCTGCTGTTTCCAGCATAGCTCTTAAAC, IDT HPLC purified).

## Analysis of CRISPR screens

Counts were determined for exact matches to guide pairs using pyCROQUET v1.6.0 (https://github.com/cancerit/pycroquet) and guides with less than 10 counts were removed based on the bimodal distribution of the library reads, and an increased coefficient of var-iation at these low read counts (Supplementary Fig. 1B). To prevent inequalities due to library size and count distribution we performed sum normalisation and scaled the resulting values by a factor of 10e6. Finally we calculate fold changes of 14 days experiment versus 3 days or the plasmid (when applicable) and log-transform them. Single guide values were aggregated at gene level by averaging their log fold change for the comparison with project SCORE single KO (Figs. 2B and S3C) and positional effect estimation (Figs. 2C and S2D). No copy number correction was applied as log fold changes were highly comparable with and without. Genetic interactions were assessed using Gemini[32] v1.5.1 (https://github.com/sellerslab/gemini). Plots were generated in R or Python.

## Co-competition assay

A co-competition assay was used to investigate the relationship between the transduction of two pairs of paralog genes (*CNOT7*-*CNOT8* and *ASF1A*-*ASF1B* – double), either simultaneously (double-transduced population) or paired with a non-essential gene (*CYP2A13*; single transduced population). Two gRNAs were used per paralog gene: CNOT7 (gRNA1: GAGGGAGGACATGTATG; gRNA2: GTAG-TAGCTCTATAGAG); *CNOT8* (gRNA1: GGATTCTCGTTTGCCAG; gRNA2: GCTGCTTATGACATCAG); *ASF1A* (gRNA1: GAGGCTGATGCACC-TAATCC; gRNA2: GATCACCTTCGAGTGCATCG); *ASF1B* (gRNA1: GCGGGTTCTCACGCAGCTCA; gRNA2: GGTTGACGTAGTAGCCCACT) were cloned as above by Gibson assembly. 1×10$^6$ HT-29 Cas9-expres-sing cells (per well) were seeded into a 6-well plate, and subsequently transduced with lentiviruses expressing each gRNA in a mCherry (Addgene #67977, *CNOT7* and *ASF1A*) or Azami Green vector (Addgene #67976, *CNOT7* and *ASF1B*). An experiment was conducted with either both gRNA1 or gRNA2 for each pair. In addition, each gene was also paired with a non-essential gene (*CYP2A13*; gRNA: GGTCACCGTGCGTGCCC). Cells were transfected with lentivirus expressing each gRNA in a mCherry (*CNOT7* or *ASF1A*) or Azami Green vector (*CNOT8* or *ASF1B*) to create four populations (non-transduced, two single transduced, one double transduced). Populations were monitored over time, and measurements for the relative depletion of the single and double-transfected populations were taken by FACS at days 3 and 14. Results were obtained by comparing the expected

phenotype of the double-transduced population (sum of the phenotypes of the single-transduced population) and the observed phenotype of double-transduced population. A double transduction of the non-essential gene was used as a negative control. All analyses were carried out using FCS Express (version 7.18.0025).

## Growth Assay

gRNA1 of each gene of a paralog pair (*CNOT7-CNOT8*, *ASF1A-ASF1B*) were cloned simultaneously as guide pairs into the pKLV2-U6gRNA5(BbsI)PGKpuro-2A-BFP-W (Addgene #67974) vector as above. In addition, each gene was also individually paired with a non-essential gene (*CYP2A13*), and a positive control (*PLK1-CYP2A13*; *PLK1* gRNA: GGCGGACGCGGACACCA) was also included. A lentivirus for each construct was used to transduce HT-29 Cas9-expressing cells ($1 \times 10^6$ cells/well in a 6-well plate). 48 h post-transduction cells were seeded into 96-well plates ($1 \times 10^3$ cells/well, 6 replicates per condition) and Puromycin selection (20 µg/ml) was carried out. Plates were loaded into an Incucyte platform and growth rate of each condition was monitored for the following 6 days. CellTiter-Glo measurements were carried out at the end of the experiment. Results were obtained by comparing the growth rate and CellTiter-Glo measurement between the cells transduced with both genes of the paralog pair vs one gene and the non-essential gene. All analysis were carried out using GraphPad Prism (version 10.1.0).

## Western Blot

Protein lysate of dual gRNA transduced cells (growth assay experiment) was collected at Day 4 post-transduction for a total of 4 samples (polybrene control, two genes of a paralog pair, each individual gene of a paralog pair with a non-essential). Knockout of either gene (when paired with non-essential) or both genes (when paired together) was confirmed by western blotting. Antibodies used were as follow: β-actin (Cell Signalling 4970 L, 1:1000 dilution); CNOT7 (abcam ab195587, 1:1000 dilution); CNOT8 (amsbio AMS.E-AB-62996-60, 1:1000 dilution); ASF1A (ProteinTech 22259-1-AP, 1:1000 dilution); ASF1B (ProteinTech 22258-1-AP, 1:1000 dilution); ECL Anti-rabbit IgG HRP-linked whole antibody (Amersham NA934, 1:2000 dilution).

## Statistics and reproducibility

No statistical method was used to predetermine sample size, and no data was excluded from the analyses. Screens were all performed in biological triplicate across at least three guide RNA pairs per target pair, which based on analysis of CRISPR screening data[1] gave optimal performance. Randomization and blinding was not necessary as experiments were performed as a pool.

## Reporting summary

Further information on research design is available in the Nature Portfolio Reporting Summary linked to this article.

# Data availability

The library design and screening analysis data have been deposited in the Zenodo database with the identifier https://doi.org/10.5281/zenodo.17191951[44]. The raw screen data is available in the Figshare database with the identifier https://doi.org/10.6084/m9.figshare.25533091.v1. The raw sequencing data is available in the EBI European Nucleotide Archive accession number ERP183979 and are available at the following URL https://www.ebi.ac.uk/ena/browser/view/ERP183979. The editing efficiency and validation data generated in this study are provided in the Supplementary Information and Source data file. Source data are provided with this paper.

# Code availability

The code used to generate analysis has been deposited in the Zenodo database (https://doi.org/10.5281/zenodo.17191951)[44,45] under a Creative

 Pilot library design: https://github.com/EmanuelGoncalves/crispy/blob/master/notebooks/dualguide/Library2Composition.py Genetic interaction library design: https://github.com/ibarrioh/DualGuide_COLO1/ Screen analysis: https://github.com/ibarrioh/DualGuide_COLO1/.

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

## Acknowledgements

We would like to thank the flow cytometry facility in Sanger Cellular Operations for analysis support and DNA sequencing facility in Sanger Scientific Operations for library construction and sequencing. pSpCas9(BB)–2A-GFP (PX458) was a gift from Feng Zhang (Addgene plasmid # 48138; http://n2t.net/addgene:48138; RRID:Addgene_48138)[46]. pKLV2-U6gRNA5(BbsI)-PGKpuro2ABFP-W, pKLV2-U6gRNA5(BbsI)PGKpuro-2A-mCherry-W and pKLV2-U6gRNA5(BbsI)PGKpuro-2A-mAG-W were gifts from Kosuke Yusa (Addgene plasmid # 67974, #67976, #67977; http://n2t.net/addgene:67974 http://n2t.net/addgene:67976 http://n2t.net/addgene:67977; RRID:Addgene_67974 RRID:Addgene_67976 RRID:Addgene_67977)[24]. We are grateful to Fiona Behan for advice and contributions to experimental design, Russell Walton for helpful comments on the manuscript and members of the Garnett and Bassett labs for their advice, discussions and support. This research was funded in part by the Wellcome Trust 220540/Z/20/A as institute core funding to M.G. and A.B. and OpenTargets OTAR2062. Funding from FCT (Fundação para a Ciência e Tecnologia), under projects UIDB/50021/2020 (DOI:10.54499/UIDB/50021/2020), https://doi.org/10.54499/2024.07252.IACDC (through RE-C05-i08.M04), and https://doi.org/10.54499/LISBOA2030-FEDER-00868200 supported E.G. For the purpose of Open Access, the author has applied a CC BY public copyright licence to any Author Accepted Manuscript version arising from this submission.

## Author contributions

A.R.B., M.J.G., T.B. and E.G. conceived and designed the study; A.R.B., M.J.G. and L-P.Z. supervised the work; T.B., L.D., R.S., E.S., M.S., A.B., G.C.C.K. performed and interpreted experiments; E.K., E.G., I.B-H. performed and interpreted computational analysis. T.B., I.B-H., E.K., E.G., A.R.B. drafted paper and all authors contributed to editing.

## Competing interests

A.B. is a founder of and consultant for Ensocell therapeutics. M.G. is a founder of and consultant for Mosaic Therapeutics, receives research funding from GSK and Astex Pharmaceuticals and is a consultant for Bristol-Myers Squibb. The remaining authors declare no competing interests.
