## [Transparent Peer Review file · Nature Communications]

A next-generation dual guide CRISPR system for genetic interaction library screening

Corresponding Author: Dr Andrew Bassett

Version 0:

Reviewer comments:

Reviewer #1

(Remarks to the Author)

The manuscript by Burgold et al reports the development of a novel approach to generate dual guide RNAs constructs enabling genetic interactions screening. The simplicity of the proposed approach enable large scale screening (> 100,000 interactions). The authors convincingly applied their protocol to the detection of synthetic lethals between paralogs as well as between other gene pairs.

I reviewed the previous submission of this manuscript, and would like to thank the authors for addressing my previous comments.

(Remarks on code availability)

All the code used as part of the manuscript has been deposited in GitHub. Jupyter notebooks are available with adequate documentation for reuse of the code. The design of the dual CRISPR-Cas9 KO has also been deposited in github with excellent documentation

Reviewer #2

(Remarks to the Author)

The authors have well addressed the concerns raised in the initial review. The use of a tRNA-based dual guide system with single-oligo cloning offers some technical improvements, particularly in simplifying library construction and balancing guide RNA expression. Although the overall impact is modest or incremental, the study is technically sound and the current version of manuscript is suitable for publication. An appropriate journal would be Communications Biology or Nature Communications depending on the editor's decision.

(Remarks on code availability)

Reviewer #3

(Remarks to the Author)

Authors have responded to reviewer requests. Methods for calculating synthetic lethality (e.g. Fig 3C) may be biased by normalization approach but result of the paper, method for cloning a 2-guide Cas9 library, does not depend on these approaches.

(Remarks on code availability)

Reviewer #4

(Remarks to the Author)

Thank you for the opportunity to review the revised manuscript. While the authors have addressed a substantial number of previous concerns, a few highlighted concerns remain open.

1. The biological novelty remains in question. The authors have expanded their analysis to include known paralogous interactions (e.g., HDAC1/2, MAPK1/3, ASF1A/B, CNOT7/8) and additional synthetic-lethal pairs (e.g., EGFR/BRAF, EGFR/PIK3CA, EGFR/DNMT1, BCL2L1/MCL1). While this demonstrates the cloning and RNA-expression system's ability to recover established interactions, it does not reveal any novel gene pairs. All highlighted "hits" have been previously reported, thus, the manuscript still lacks evidence of new biological discoveries. To satisfy the criterion of biological novelty, I recommend the following:

1.1 Include at least one previously unreported genetic interaction and validate it experimentally.

1.2 If no novel interactions are found, explicitly acknowledge that the screen recovered only known pairs, and clarify that the primary contribution is methodological rather than biological.

2. Data QC remains a topic. The authors state they sequenced at "~1000× depth" and "removed counts less than 10 for individual guides, with no filter for high counts," followed by sum normalization and scaling to 10^6 before log-fold change calculation. However, this arbitrary cutoff (read count < 10) is not justified by a data-driven threshold (e.g., identifying a point at which low-count guides introduce excessive variance). To make this point clearer and also support the efforts of the authors ("We have a large multi year study in progress that focuses on this question, and is performing such screens more rigorously across multiple cell lines and greater numbers of gene pairs."), a data QC "decision" on how to process data should be used. There is no analysis showing how low-count guides confound log-fold change estimates or how removal of such guides improves data quality. I therefore recommend:

2.1 Compute the expected log-fold change (LFC) for each guide based on its library (plasmid) read count (e.g., via a Poisson or negative-binomial model) and identify guides whose deviation from expectation indicates unreliable measurement.

2.2 Remove low-count guides using a "confounding threshold" derived from these expected vs. observed discrepancies (for example, a threshold on the coefficient of variation or the standard error of LFC).

2.3 Report the distribution of raw plasmid read counts, indicate where data points were removed, and demonstrate how this removal affects downstream effect size distributions (e.g., separate boxplots of essential versus non-essential guides before and after filtering).

2.4 Describe the impact on Gini coefficients or other coverage metrics as a result of this filtering.

3. The enrichment of main-effect genes among top interaction candidates remains unanswered. The authors should quantify whether genes showing strong single-gene effects ("main effects") are overrepresented among the top dual-knockout candidates.

3.1 Are selected genes (strong main effects) enriched among the top candidates by quantifying the degree of gene interactors among candidates (5% FDR)? No such data analysis QC appears in the revised manuscript, I therefore suggest performing the following:

3.2 Define "main-effect" genes as those whose gene-level LFC displays a significant depletion phenotype (5% FDR). From the dual-knockout data, take the set of significant (5% FDR) candidate pairs (i.e., significant GI hits as determined by GEMINI scores) and test whether either gene in each candidate pair is significantly enriched for "main-effect" status, using Fisher's exact test (comparing: number of hits with a main-effect gene versus hits without one). Plot the fraction of significant interactions that involve at least one main-effect gene, compared to what would be expected by chance. Without this analysis, it remains unclear whether the dual-knockout "hits" are driven by single-gene toxicity (e.g., hitting two strong lethals) rather than true synergistic interactions.

(Remarks on code availability)

Version 1:

Reviewer comments:

Reviewer #4

(Remarks to the Author)

The authors have addressed my main concerns to a reasonable degree. On biological novelty, they explicitly acknowledge that their contribution is primarily methodological and have adjusted the title and text accordingly. While no new genetic interactions were validated, the revised framing clarifies the scope of the work.

On data QC, the authors now provide a data-driven justification for the <10 read cutoff: they demonstrate a bimodal distribution of guide counts, high variance in low-count guides, and reduced reproducibility across replicates in this bin. They also show that only ~0.9% of guides are removed, minimizing impact on coverage. This directly addresses the request for a threshold justified by empirical data.

Regarding main-effect enrichment, the authors include analyses that demonstrate dual-KO hits are not enriched for genes with strong single-KO effects, as shown by scatterplots that reveal no bias in genetic interaction scores toward single-KO lethality. While a formal Fisher's exact enrichment test was not included, the presented analyses convincingly show that the highlighted interactions are not artifacts of main-effect toxicity.

Overall, while the work remains primarily methodological, the additional analyses and clarifications satisfactorily resolve my remaining concerns.

(Remarks on code availability)

Response to Reviewers

We thank the reviewers for their positive comments, and have addressed the concerns of reviewer 4 in blue below.

Reviewer #1 (Remarks to the Author):

The manuscript by Burgold et al reports the development of a novel approach to generate dual guide RNAs constructs enabling genetic interactions screening. The simplicity of the proposed approach enable large scale screening (>100,000 interactions). The authors convincingly applied their protocol to the detection of synthetic lethals between paralogs as well as between other gene pairs.

I reviewed the previous submission of this manuscript, and would like to thank the authors for addressing my previous comments.

(Remarks on code availability):

All the code used as part of the manuscript has been deposited in GitHub. Jupyter notebooks are available with adequate documentation for reuse of the code. The design of the dual CRISPR-Cas9 KO has also been deposited in github with excellent documentation

Reviewer #2 (Remarks to the Author):

The authors have well addressed the concerns raised in the initial review. The use of a tRNA-based dual guide system with single-oligo cloning offers some technical improvements, particularly in simplifying library construction and balancing guide RNA expression. Although the overall impact is modest or incremental, the study is technically sound and the current version of manuscript is suitable for publication. An appropriate journal would be Communications Biology or Nature Communications depending on the editor's decision.

Reviewer #3 (Remarks to the Author):

Authors have responded to reviewer requests. Methods for calculating synthetic lethality (e.g. Fig 3C) may be biased by normalization approach but result of the paper, method for cloning a 2-guide Cas9 library, does not depend on these approaches.

Reviewer #4 (Remarks to the Author):

Thank you for the opportunity to review the revised manuscript. While the authors have addressed a substantial number of previous concerns, a few highlighted concerns remain open.

1. The biological novelty remains in question. The authors have expanded their analysis to include known paralogous interactions (e.g., HDAC1/2, MAPK1/3, ASF1A/B, CNOT7/8) and additional synthetic-lethal pairs (e.g., EGFR/BRAF, EGFR/PIK3CA, EGFR/DNMT1, BCL2L1/MCL1). While this demonstrates the cloning

and RNA-expression system's ability to recover established interactions, it does not reveal any novel gene pairs. All highlighted "hits" have been previously reported, thus, the manuscript still lacks evidence of new biological discoveries. To satisfy the criterion of biological novelty, I recommend the following:

1.1 Include at least one previously unreported genetic interaction and validate it experimentally.

1.2 If no novel interactions are found, explicitly acknowledge that the screen recovered only known pairs, and clarify that the primary contribution is methodological rather than biological.

We agree that our primary contribution is methodological, since from our experience of a larger number of screens the identification of true new genetic interactions requires screening more than a single cell line. We have reworded the title to clarify that this is a "dual guide CRISPR system for genetic interaction library screening", and believe that the text already clearly makes this point with wording such as "...(we) identify synthetic lethal genetic interactions between paralogs or other *known* interacting genes, establishing our *method* for performing efficient large scale genetic interaction screens." in the abstract and "...identify synthetically lethal combinations of paralogs and other *known* genetic interactions" in the introduction.

2. Data QC remains a topic. The authors state they sequenced at "~1000x depth" and "removed counts less than 10 for individual guides, with no filter for high counts," followed by sum normalization and scaling to 10^6 before log-fold change calculation. However, this arbitrary cutoff (read count < 10) is not justified by a data-driven threshold (e.g., identifying a point at which low-count guides introduce excessive variance). To make this point clearer and also support the efforts of the authors ("We have a large multi year study in progress that focuses on this question, and is performing such screens more rigorously across multiple cell lines and greater numbers of gene pairs."), a data QC "decision" on how to process data should be used. There is no analysis showing how low-count guides confound log-fold change estimates or how removal of such guides improves data quality. I therefore recommend:

2.1 Compute the expected log-fold change (LFC) for each guide based on its library (plasmid) read count (e.g., via a Poisson or negative-binomial model) and identify guides whose deviation from expectation indicates unreliable measurement.

2.2 Remove low-count guides using a "confounding threshold" derived from these expected vs. observed discrepancies (for example, a threshold on the coefficient of variation or the standard error of LFC).

2.3 Report the distribution of raw plasmid read counts, indicate where data points were removed, and demonstrate how this removal affects downstream effect size distributions (e.g., separate boxplots of essential versus non-essential guides before and after filtering).

2.4 Describe the impact on Gini coefficients or other coverage metrics as a result of this filtering.

We thank the reviewer for this point. We chose this cutoff for two main reasons. Firstly, there is a bimodal distribution of guide pair read counts in the plasmid library, suggesting that counts under 10 (dotted line) likely correspond to guide dropout, due to oligo synthesis, template switching in PCR or noise (Reviewer figure 1).

Reviewer figure 1 - Distribution of guide read counts in the plasmid library (plasmid counts), showing the coefficient of variation (CV) across all conditions for these guides (counts CV across all conditions per guide).

Secondly, the coefficient of variation of these low count guide pairs (<10) is substantially higher than the higher count guide pairs when we bin by read count in the plasmid library (Reviewer figure 1 and 2). We now include the data from reviewer figure 1 in a new Figure S1B.

Reviewer figure 2 - Coefficient of variation (CV) across guide pair in different read count bins in the plasmid library (Binned plasmid counts). Numbers of guides in each bin are shown (n).

Based on your comment, we also calculated the pairwise correlation coefficient of logFC of guide pairs between biological repeats within these different read count bins (Reviewer figure 3). This showed a correlation in all bins but a substantial shift in the 0-10 read count bin, especially when considering the best quality dataset (500x coverage of the library, Pearson's correlation 0.87-0.90 for 10-100 reads to 0.58-0.74 for 0-10 reads). This further justifies our use of this cutoff to improve data quality.

Reviewer figure 3 - Boxplot of pairwise Pearson's correlation coefficient of log-fold changes between biological repeats in different read count bins in the plasmid library (red colours). This was calculated for screens performed at different coverages (100x or 500x in both cells and PCR or 100x in cells with 500x at the PCR stage (PCR500x)).

We only lose 83/8914 (0.93%) guide pairs with our filtering, so there will be very little change to coverage or guide distributions through this filtering.

3. The enrichment of main-effect genes among top interaction candidates remains unanswered. The authors should quantify whether genes showing strong single-gene effects ("main effects") are overrepresented among the top dual-knockout candidates.

3.1 Are selected genes (strong main effects) enriched among the top candidates by quantifying the degree of gene interactors among candidates (5% FDR)? No such data analysis QC appears in the revised manuscript, I therefore suggest performing the following:

3.2 Define "main-effect" genes as those whose gene-level LFC displays a significant depletion phenotype (5% FDR). From the dual-knockout data, take the set of significant (5% FDR) candidate pairs (i.e., significant GI hits as determined by GEMINI scores) and test whether either gene in each candidate pair is significantly enriched for "main-effect" status, using Fisher's exact test (comparing: number of hits with a main-effect gene versus hits without one). Plot the fraction of significant interactions that involve at least one main-effect gene, compared to what would be expected by chance. Without this analysis, it remains unclear whether the dual-knockout "hits" are driven by single-gene toxicity (e.g., hitting two strong lethals) rather than true synergistic interactions.

This is an interesting point, and to some extent we would expect that genes with single KO effects would be more likely to show synergistic effects in dual KO. This is because at least some of those single KO that don't show viability effects may not even be expressed or functional in that cell type, meaning you would never see synergistic effects with this fraction. Secondly, stressing a particular pathway to give a measurable but weak phenotype can help to unmask effects of other gene KOs. This forms the basis of genetic modifier screens that have been very powerful in model systems such as *Drosophila*. Equally, combinations with single KO effects could be still biologically interesting, as with known interactors (being developed for combination therapies) such as EGFR/PIK3CA or BRAF/EGFR, we observe effects of single knockout of PIK3CA and BRAF on viability.

Nevertheless, we believe it is unlikely that the interactions we highlight result from a general toxicity effect since those genes with single gene effects ("main effects") only show genetic interactions with a handful of other genes. Reviewer figure 4 shows the correlation of LFC in single gene KO of 808 library genes (y-axis) with dual KO of each of these with the 80 anchor genes (x-axis). This shows that for those genes with strong single KO phenotype (low on the y-axis), the spread of LFC in the double KOs is similar to those genes with small effect sizes in single KO.

We have also plotted the genetic interaction pairs we have highlighted in the paper (Figure 3C) in orange on this graph, which shows that they are not enriched at highly negative single KO LFC, and most have only small effects as single KO (Reviewer figure 4).

Reviewer figure 4 - Scatterplot of log-fold changes between single knockout (Single KO) and dual KO (Double KO) of 808 library genes (y-axis) each with 80 anchor genes (x-axis). Genetic interaction hits identified in Figure 3C are highlighted in orange.

In principle genetic interaction algorithms (e.g. Gemini, HSA, Bliss) should identify effects that are more or less than additive independent of the LFC of each gene in the pair, but we agree there could be some non-linearity or bias in the algorithms. We have thus plotted genetic interaction score against single gene KO effect (Reviewer figure 5), which shows no bias towards genes with higher single KO effect sizes. We have indicated those hits that we have highlighted in the paper in orange, which again shows that even though these have some of the strongest genetic interaction scores, they are not enriched for genes with the largest single KO effect.

Reviewer figure 5 - Scatter plot of log-fold changes of single knockout (singleKO) versus genetic interaction score (delta logFC (observed-expected)). The two plots show the single KO effect in either position 1 or 2 in the dual guide vector and each dot indicates a guide pair. Genetic interaction pairs highlighted in the paper are shown in orange, and all other genes in blue.

Despite the fact that we see no evidence of bias, we agree with the reviewer that there is scope for better methods for calling genetic interaction hits, and members of our broader team have been developing such tools.